# A neural theory for counting memories

Sanjoy Dasgupta[1], Daisuke Hattori [2] ✉ & Saket Navlakha [3] ✉

Keeping track of the number of times different stimuli have been experienced is a critical computation for behavior. Here, we propose a theoretical two-layer neural circuit that stores counts of stimulus occurrence frequencies. This circuit implements a data structure, called a *count sketch*, that is commonly used in computer science to maintain item frequencies in streaming data. Our first model implements a count sketch using Hebbian synapses and outputs stimulus-specific frequencies. Our second model uses anti-Hebbian plasticity and only tracks frequencies within four count categories ("1-2-3-many"), which trades-off the number of categories that need to be distinguished with the potential ethological value of those categories. We show how both models can robustly track stimulus occurrence frequencies, thus expanding the traditional novelty-familiarity memory axis from binary to discrete with more than two possible values. Finally, we show that an implementation of the "1-2-3-many" count sketch exists in the insect mushroom body.

"I've never smelled anything like this." "I've seen you once before." "I've heard this song many times." Estimating the frequencies of different stimuli experienced is an important computation that requires storing and updating the number of times each stimulus has been observed. This computation occurs ubiquitously across sensory modalities, and naturally without reward or punishment, allowing organisms to make rapid behavioral decisions, absent any specific details about the memory[1].

One line of evidence that the brain keeps track of stimulus occurrence frequencies comes from studies of recognition memory[2], which report neurons whose activity encodes whether a stimulus is novel or familiar. Recognition memory exists for many types of stimuli, including visual[3,4], auditory[5–7], and olfactory[8,9]. Most studies report neurons whose response magnitudes decrease with familiarity; i.e., neurons show strong responses upon the first presentation of the stimulus, and weaker responses to subsequent presentations (called repetition suppression[10]). Others have found neurons that become more active with familiarity (called repetition enhancement[11,12]). While many computational models of recognition memory have been proposed[13–17] (see review by Bogacz and Brown[18]), most models consider familiarity discrimination as a binary problem — is the stimulus novel or familiar? — as opposed to a problem where the desired output is an estimate of how many times the stimulus has been experienced. In addition, classical models are not well integrated with modern

experimental data revealing how neural circuits represent stimuli in high-dimensional spaces and update their frequencies at synaptic resolution.

Frequency estimation is distinct from the numbers sense[19–21], which underlies the ability to perform approximate numerical comparisons. For example, when frogs chose between patches of food items, their choice between three and four items is random, but they reliably chose six items over three[20]. Similar behaviors have been observed across the animal kingdom[22] — including in primates[23,24], reptiles[25], fish[26–28], birds[29], flies[30], and bees[31–33] — without relying on language or numerical symbols. While useful for quantifying magnitudes — the number of food items in a patch, the number of predators in a group — the numbers sense does not provide a way to store a mapping from observed items to frequency counts, nor a way to update counts as items are experienced over time.

In computer science, frequency estimation comes up in many applications, such as keeping track of the number of times different videos are watched or different songs are played, to identify popular content. This problem is commonly solved using a data structure called a count sketch[34,35]. Much like how an artistic sketch provides a quick approximation of a complex drawing, a "sketch" is a data structure that provides approximate answers to a query, while consuming substantially (often exponentially) less space than what would be required to store all of the data. A "count sketch" is a sketch that supports the frequency

[1]Computer Science and Engineering Department, University of California San Diego, La Jolla, CA 92037, USA. [2]Department of Physiology, UT Southwestern Medical Center, Dallas, TX 75390, USA. [3]Simons Center for Quantitative Biology, Cold Spring Harbor Laboratory, Cold Spring Harbor, NY 11724, USA. ✉e-mail: daisuke.hattori@utsouthwestern.edu; navlakha@cshl.edu

estimation query; i.e., "how many times have I seen item $x$?". Count sketches are primarily used in instances where large amounts of data are continuously processed and where storing all of the data is prohibitive.

Here, we develop a theory for keeping track of stimulus occurrence frequencies, while being tolerant to noise. Our proposed neural circuit implements a count sketch using a two-layer neural architecture: a sparse, high-dimensional stimulus encoding layer that synapses onto a decoding layer with one neuron, which outputs the frequency of any stimulus. We also propose a variant of the model, called the "1-2-3-many" sketch, that only tracks frequencies within four categories, ranging from novel (frequency = 1) to very familiar (frequency > 3). Both models effectively expand the classic novelty-familiarity axis from a binary state memory system to one with more than two discrete states. We empirically demonstrate the accuracy of both neural count sketches on three datasets, and we derive mathematical bounds of their error as a function of environmental and neural variables (e.g., number of stimuli observed, number of encoding neurons, synaptic precision). Finally, we show that all the circuitry needed to implement the "1-2-3-many" count sketch − including network architecture, synaptic plasticity rule, and output neuron that encodes count categories − exists in the insect mushroom body, and re-analysis of published experimental data indeed shows that novelty responses can be distinguished along the four categories proposed. We conclude by raising several testable experimental hypotheses, and by describing other brain regions that have all the machinery needed to support memory counting.

## Results

We begin by presenting the count sketch data structure as a solution to the memory counting problem. We then present a neural implementation of the count sketch and show that it works well in practice and in theory. Finally, we show that three main requirements of our model − the circuit architecture, the synaptic plasticity rule induced after stimulus observation, and the response precision of the counting neuron − exist in the insect mushroom body.

### The count sketch data structure for frequency estimation in streaming data

Say we are given a sequence of observed items, where each item is drawn from a set $\mathcal{X} = \{x_0, x_1, \ldots, x_N\}$ of $N$ possible items. The sequence can contain the same item multiple times, and we would like to keep track of the number of times each unique item is seen. A hash table mapping keys (items) to values (counts) would provide exact counts but would require storing each item in its entirety, which would be costly if the items are large (e.g., videos or songs) and numerous. A *count sketch* is a data structure that outputs counts for an item that are approximately equal to the true counts of the item, while only requiring a few bits of storage space per item, no matter how big the items themselves are.

A count sketch stores a frequency table for items using a 2D matrix $C$ with $k$ rows and $v$ columns, where $k$ is the number of hash functions, and $v$ is the range of the hash functions (Fig. 1A). Each row is associated with a hash function $h : x \to [v]$; i.e., the function takes as input some item $x$ and maps it to a column index in $C$. The $k$ hash functions are pairwise independent and random. This means that the inputs are spread uniformly over the range, and two similar inputs could be assigned to arbitrarily far apart indices. In Fig. 1A, there are three hash functions ($k = 3$). Each entry in $C$ corresponds to a counter and is initialized to 0.

To insert an item $x$ into the count sketch, for each hash function $i$, we compute $j = h_i(x)$, and then we increment $C[i, j]$ by 1. In Fig. 1A, $h_1(x_1) = 1$, which means that the first hash function maps input $x_1$ to column 1. So, when $x_1$ is observed (Fig. 1C, left), we increment $C[1, 1]$ by 1. Similarly, $h_2(x_1) = 2$, which means we increment $C[2, 2]$ by 1, and $h_3(x_1) = 5$, which means we increment $C[3, 5]$ by 1. After these three

entries are modified, we are finished inserting $x_1$. This process repeats for each subsequent item (Fig. 1C, right).

At any point, we can query the count sketch for the estimated frequency of item $x$ (Fig. 1D):

$$\hat{f}(x) = \frac{1}{k} \sum_i C[i, h_i(x)].$$

Intuitively, each row stores a predicted count for the item using a single hash function, which is then aggregated (averaged) over the rows into a final estimate. Other aggregate functions[36] include median[34] and min[35,37], which have also been implemented in spiking neural networks[38].

The accuracy of the estimate depends on the values chosen for $k$ (the number of rows) and $v$ (the number of columns). If $v$ is large enough such that each unique item observed is mapped to a unique column index, then only a single row ($k = 1$) is needed to generate exact count estimates. However, in practice, hash collisions (overlaps) are likely, where a hash function maps two different items to the same column index. For example, in Fig. 1D, the counts for $x_1$ and $x_2$ are exactly correct because each item is mapped to a unique set of column indices that do not overlap with those of other observed items. On the other hand, despite $x_3$ never being observed in the input sequence, the count sketch would estimate its frequency to be 1/3 because $h_2$ maps both $x_3$ and $x_2$ to the same column index (3). Thus, the level of approximation (i.e., the amount of deviation from the correct count) depends on the amount of overlap with other items, as well as the number of rows that are averaged over. Overall, larger values of $k$ and $v$ provide more accurate estimates, at the expense of larger space consumption. Typically, $v$ is set much larger than $k$ since $v$ relates to the error of the count estimate for each hash function, and $k$ simply averages these errors over multiple, independent hash functions.

### A neural implementation of a count sketch

There is a very simple way that neural circuits can implement a count sketch data structure (Fig. 1B). The main idea is to "flatten" the 2D matrix of counters with $k$ rows and $v$ columns into a 1D array of $k \times v$ synapses. In the count sketch, each input modifies the values of $k$ entries in the matrix (one per row). In the neural version, each input will modify $k$ synaptic weights. The identity of these $k$ synapses will be determined by a neural hash function, which will encode inputs using sparse, high-dimensional representations. Specifically, of the $k \times v$ presynaptic neurons, only $k \ll v$ will fire per input, and the synapses of these neurons are modified for the input. Post-synaptically, there is one decoding neuron that reads-out from the encoding neurons and outputs a frequency for the given stimulus.

These three pieces (stimulus encoding, synapse weight updating, and frequency decoding) are described below.

**Stimulus encoding.** The first piece determines which pre-synaptic neurons are active for an input. This requires designing a neural hash function, $h : \mathcal{R}^d \to \{0, 1\}^m$, which takes some input vector $x \in \mathcal{R}^d$ and assigns it to a point in $m$-dimensional space, where $m = kv$. A canonical way to do this is via random projection and sparsification[39]. This motif is used widely, including in the olfactory system[40–42], hippocampus[43], and cerebellum[44], to create sparse, high-dimensional representations for inputs[45,46].

In the random projection step, we compute $y = (y_1, y_2, \ldots, y_m) \in \mathcal{R}^m$ by:

$$y = Mx,$$

where $M$ is a random matrix of size $m \times d$. For example, $M$ can be a Gaussian random matrix, where each value is drawn i.i.d. from $\mathcal{N}(0, 1)$;

**Fig. 1 | The count sketch and corresponding neural circuit implementation.**
**A** The count sketch data structure is a 2D matrix $C$ of counters with $k$ rows and $v$ columns. There is one hash function $h$ per row, each of which determines which column in the row is modified when an item $x$ is observed (dotted arrows). **B** The neural implementation of a count sketch uses a 1D array $w$ of $k \times v$ synapses. When an item is observed, the synapses of the $k$ pre-synaptic neurons that are active for the item (orange highlight) are modified. The neurons activated for the item are determined using a hash function that assigns the item a sparse, high-dimensional representation ($z$). In this example, each item $x$ is a $d$-dimensional vector. **C** To insert an item from the sequence into the count sketch (top), $k$ counters are incremented. For example, after the first time ($x_1$) is observed, in the first row, the

counter in the 1st column (i.e., $C[1,1]$) is incremented by 1 since $h_1(x_1) = 1$. In the second row, $C[2,2]$ is incremented by 1 since $h_2(x_1) = 2$. In the third row, $C[3,5]$ is incremented by 1 since $h_3(x_1) = 5$. Similarly, in the Hebbian neural count sketch (bottom), the synaptic weights of the $k$ activated pre-synaptic neurons for $x_1$ (orange highlight) are incremented. In the anti-Hebbian model, all synaptic weights are initialized to 1, and synapses active for the item are decremented with each observation. **D** To output an estimate of the frequency of item $x_1$, the count sketch computes: $(1/k) \sum_{i=1}^{k} C[i, h_i(x_1)]$ — i.e., the average of the predicted counts over the rows. This results in correct estimates for $x_1$ and $x_2$, and a near-correct estimate for $x_3$. Similarly, the neural count sketch outputs: $(1/k) \sum_{i, z_i > 0} w_i z_i$ — i.e., the average of the weights of the activated neurons for the item.

or, it could be a sparse binary matrix, where each row of $M$ has a small number of 1s and the rest of the values are 0.

In the sparsification step, we compute $z = (z_1, z_2, ..., z_m) \in \{0, 1\}^m$, where:

$$z_i = \begin{cases} 1 & \text{if } y_i \text{ is one of the } k \text{ largest entries of } y \\ 0 & \text{otherwise.} \end{cases}$$

In other words, only the $k$ neurons that fire at the highest rate among the population remain firing, and the rest are silenced. Mechanistically, this is implemented by inhibitory neurons, which receive excitatory input from the encoding neurons and provide feedback inhibition, which silences all except the highest firing neurons. This computation is often dubbed a "$k$-winners-take-all" competition[47–49].

Importantly, unlike the random hash functions typically used in count sketches, where a small change in the input could result in an

arbitrarily far apart representation, this neural hash function is locality-sensitive[50–53]. This means that the more similar two inputs are, the more overlap there will be in their sparse representations. Biologically, this property is useful because it allows count estimates to be noise-tolerant[1]. In other words, instead of counting the frequency of $x \in \mathcal{R}^d$, we want to count the total frequency of all items within a small radius around $x$, where the radius encapsulates noisy observations of $x$.

**Synapse weight updating.** The second piece involves modifying the synaptic weights $w = (w_1, w_2, ..., w_m)$ of the $m$ encoding neurons each time an input is observed. To mimic the way counters are updated in the count sketch, all weights are initialized to 0, and the update rule is:

$$w_i = \begin{cases} w_i + 1 & \text{if } z_i > 0 \\ w_i - \epsilon & \text{otherwise.} \end{cases} \tag{1}$$

In other words, $w_i$ increases by 1 if $z_i$ is active for the input, and otherwise, $w_i$ remains the same, modulo a small memory decay parameter $\epsilon$ (in our experiments, we set $\epsilon = 0$). This is effectively a Hebbian model (i.e., repetition enhancement) and leads to neurons whose activity scales with stimulus familiarity.

**Frequency decoding.** The third piece involves a read-out neuron, which outputs stimulus-specific frequencies. For a given input $x$, this neuron computes:

$$\hat{f}(x) = \frac{1}{k}\sum_{i=1}^{m} w_i z_i,$$

that is, the average of the $k$ synapses activated for $x$, which is an estimate of the count of $x$. Since it may not be possible for a neuron to compute the average of its inputs, a simple alternative is to change the weight update in Eq. (1) to $w_i = w_i + 1/k$, and then the decoder only needs to take the weighted sum of its inputs.

Thus, a fundamental counting data structure has a simple neural correlate.

### Deriving a "1-2-3-many" count sketch
While the neural circuit described above implements a count sketch data structure, there are several problems with this model in terms of neural plausibility. First, in computer science, count sketches are primarily designed to identify "heavy hitters" − i.e., very popular items, such as videos that are watched many times − with less precision in the counts of rare items. However, biologically, "light hitters", such as items never seen before or just seen once or twice, are critical to distinguish because they signify novelty and degrees of familiarity. Second, behaviorally, the granularity of counts is likely not very high; e.g., it may not be possible (or even valuable) for organisms to distinguish between items seen 47 vs. 48 times, or between items seen 47 vs. 59 times. This is due to limits in the number of discrete firing rates that can be interpreted downstream as distinct, and limits in synaptic precision[54]. Third, experimental evidence suggests that recognition memory is largely based on repetition suppression[8–10,55–59], as opposed to repetition enhancement.

To address these issues, we propose a "1-2-3-many" sketch, that only distinguishes amongst four categories of counts:

- '1': novel (first experience).
- '2': weakly familiar (more than just one random experience).
- '3': moderately familiar
- 'many': strongly familiar (constantly re-occurring experiences)

We hypothesize that these four categories provide the best "bang for the buck", in terms of ethological value to survival and precision to encode, with larger counts having increasingly diminishing returns. Novel items (category 1) are clearly important, as they alert organisms to new and potentially salient events[60]. However, many stimuli are experienced once randomly, without much significance, and only a fraction of these stimuli are experienced twice (category 2). The two latter categories further separate environmental patterns from environmental stochasticity (Discussion). Thus, associating stimuli with graded levels of familiarity[55,61,62] could increase the behavioral repertoire of organisms.

How can we devise a 1-2-3-many sketch? The only change required is in the weight update rule. Previously, we initialized weights to 0 and applied a Hebbian update. Here, we initialize weights to 1 and apply an anti-Hebbian update, with the following functional form:

$$w_i = \begin{cases} w_i e^{-\beta} & \text{if } z_i > 0 \\ w_i + \epsilon & \text{otherwise}. \end{cases} \quad (2)$$

In other words, the weight is roughly 1 if the item is being experienced for the first time; $e^{-\beta}$ for the second experience; $e^{-2\beta}$ for the third experience; and less than $e^{-3\beta}$ for all subsequent experiences.

Thus, novel items have large responses, which decrease multiplicatively with familiarity[56,63], and the decoder neuron only needs to have four distinct responses, each representing a count category. Compared to the Hebbian model, this model creates greater separation between count categories, which makes it easier to read-out and control behavior (Discussion), at the expense of encoding fewer categories. In addition, all weights will be bounded between 0 and 1 (assuming $\epsilon = 0$; otherwise, saturation can clip weights at 1).

### The neural count sketches accurately track item frequencies in streaming data
We tested the accuracy of count estimates from the two neural count sketches using streaming data from synthetic and real-world datasets, to demonstrate how well they work in practice.

**Datasets and experimental setup.** The first dataset, Synthetic, consists of $N = 1000$ items with $d = 50$ dimensions per item, where each dimension is drawn randomly from an exponential distribution. This distribution was selected because several types of neural stimuli, such as faces[64] and odors[48], are encoded as an exponential distribution of firing rates over a population of neurons. The second dataset, Odors, is experimentally collected response data of $d = 24$ olfactory receptor neurons in the fruit fly to $N = 110$ odors[65]. The third dataset, MNIST, consists of $N = 10,000$ images of handwritten digits, where each image is of dimension $d = 84$ (after applying a pre-processing step to extract discriminative features; Supplementary Methods). We reduced each dataset such that there were no pairs of items that were very highly correlated (Pearson $r \geq 0.80$). We did this because correlated items have highly overlapping representations and thus counts that would interfere with each other; moreover, such pairs of stimuli may be difficult for animals to distinguish without training. Nonetheless, many pairs of moderately correlated items were retained. For all datasets, we set $m = 10,000$ (number of encoding neurons) and $k = 10$ (sparsity of the representation).

To generate the sequence of observed items, from each reduced dataset ($\mathcal{X}$), we drew $n$ random samples with replacement according to a Zipf (power-law) distribution. The Zipf distribution captures frequency occurrence data in many domains[66], and allows us to explore the full gamut of counts, from those items never observed in the sequence to those observed many times.

After the $n$ items were inserted into the sketch, we iterated through each unique item $x$ in the dataset and compared its ground-truth count to its predicted count, $\hat{f}(x)$, from the sketch. To test robustness to noise, we compared the ground-truth counts for $x$ to the predicted count $\hat{f}(x')$, where $x'$ is the same as $x$ but where each dimension is multiplied independently by a random value in $[0.85, 1.15]$ (i.e., up to 15% noise is added to $x$).

See Supplementary Methods for full details.

**The Hebbian neural count sketch generates signals that scale with item frequencies.** Recall that the neural count sketch uses a Hebbian learning model (i.e., repetition enhancement), and the output from the decoder neuron should correlate with the frequency of the item. This mimics neurons that become more active with familiarity.

On the Synthetic dataset, the output from the decoder neuron was highly correlated with the true count estimate ($r = 0.935$; Fig. 2A). Without noise, all count estimates are either on or above the $y = x$ line because the count sketch is a biased estimator (i.e., it can over-estimate counts, but not under-estimate). With noise added (Fig. 2B), the correlation only reduced to $r = 0.880$. Thus, count estimates for an item are robust to reasonable levels of variation in the item. This is due to the use of a locality-sensitive hash function, which ensure that very

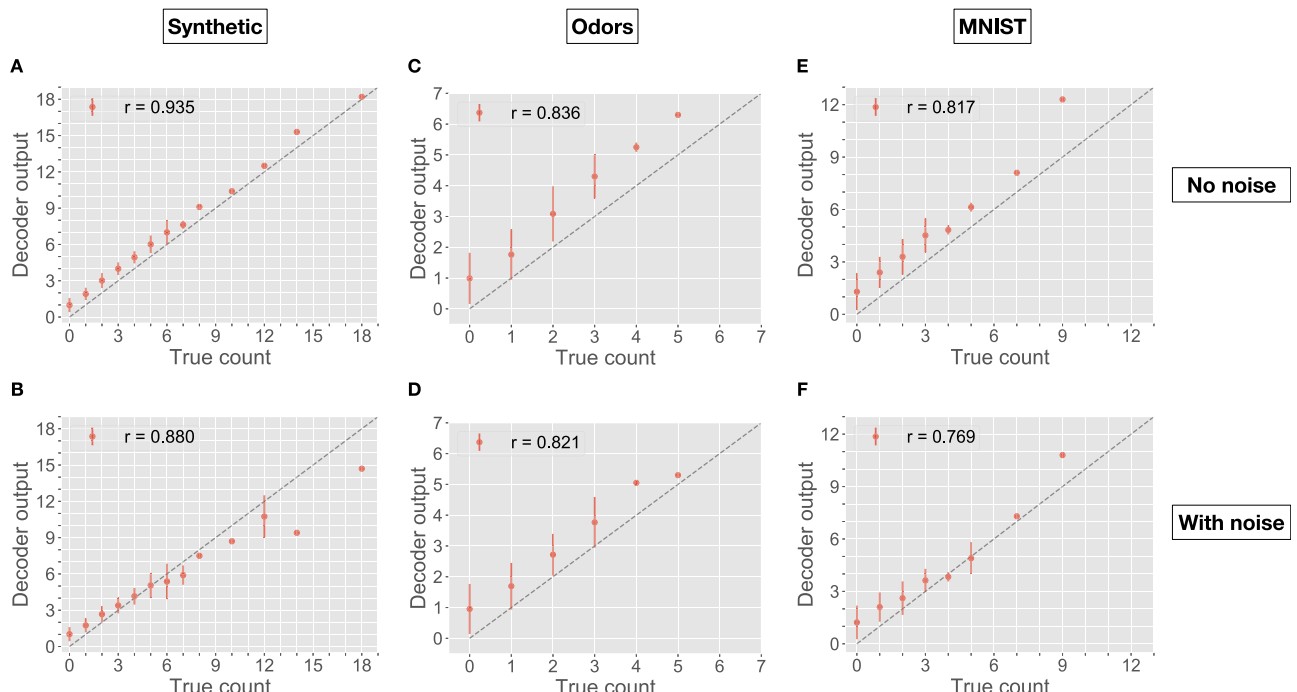

**Fig. 2 | Performance of the neural count sketch (Hebbian model).** In each panel, the x-axis shows the ground-truth count, and the y-axis shows the predicted count, as outputted from the decoder neuron. Dots show average decoder responses for items of the same ground-truth count, and error bars indicate standard deviation. Perfect performance would lie on the dotted $y = x$ line. Pearson correlation coefficient ($r$) quantifies performance accuracy (larger is better). After removing correlated items, $N = 1000$ for Synthetic; $N = 62$ for Odors; and $N = 180$ for MNIST. Each panel shows a dataset (columns), with or without noise added to items (rows). **A, B** Synthetic without and with noise. **C, D** Odors without and with noise. **E, F** MNIST without and with noise.

similar items are mapped to overlapping representations in high dimensions[50–52].

On the Odors and MNIST datasets, we observed similar trends, with a high correlation ($r = 0.836$ and $r = 0.817$; Fig. 2C, E) between ground-truth and predicted counts without noise, and with small losses in performance with noise ($r = 0.821$ and $r = 0.769$; Fig. 2D, F). Much of the error can be attributed to groups of moderately correlated items, whose counts collectively interfere with each other. For example, if we reduced the Odors dataset further by ensuring that the maximum pairwise similarity between any two items was $r = 0.70$ (instead of 0.80), then with noise, the correlation between predicted and true counts increases from 0.821 to 0.880.

Overall, the neural (Hebbian) implementation of the count sketch data structure works well in estimating counts, even for items that partially overlap.

**The anti-Hebbian neural count sketch provides a mechanism to distinguish 1-2-3-many.** Recall that the 1-2-3-many sketch uses an anti-Hebbian learning model (i.e., repetition suppression). To gauge performance of this sketch, we asked how distinguishable are the responses from the decoder neuron for items in the four count categories.

On all three datasets (Fig. 3, top), we see characteristic repetition suppression, where novel items have large decoder responses, which reduce with familiarity. For example, for the Odors dataset, items in category '1' (novel) have an average response of $0.749 \pm 0.168$, whereas items in category '2' have an average response of $0.351 \pm 0.077$, and this continues further with familiarity: $0.142 \pm 0.034$ for category '3', and $0.040 \pm 0.022$ for 'many'. All three comparisons − response magnitudes of 1-vs-2, 2-vs-3, and 3-vs-many − are significantly different ($p < 0.01$; Wilcoxon rank-sum test). With noise (Fig. 3, bottom), there is more variation as expected, but all four categories remain distinguishable.

Thus, across three diverse datasets, the 1-2-3-many sketch provides sufficient granularity to robustly categorize items into four count categories.

## Theoretical analysis of the neural count sketches

To extrapolate from the empirical results and quantify how the accuracy of count estimates depends on environmental and neural circuit variables − such as the number of stimuli observed, the number of encoding neurons, the sparsity of representations, and synaptic precision − we mathematically analyzed the neural count sketch (Hebbian model) and the 1-2-3-many sketch (anti-Hebbian model). These models have several degrees of freedom, including the length ($m$) and sparsity ($k$) of the representations and, crucially, the distribution over random matrices $M$. In Supplementary Notes 1–3, we present results of significant generality, with full proofs. Here we summarize our main results and then present a special case as an illustration.

The primary setting we consider is one in which there are $N$ distinct items (e.g., odors) that are well-separated from each other, in the sense that the distance between them is roughly what would be expected if they were chosen independently at random; this is formalized in Assumption 1. The sketching scheme is shown a sequence of $n$ observations drawn from these $N$ items, where the items are interleaved arbitrarily and might appear multiple times. Information about the observations gets coded in the weights $w_j$, and when a subsequent query $x$ (also one of the $N$ items) is made, the sketch produces a frequency estimate for it. We study how close this frequency estimate is to the actual number of times $x$ appeared in the sequence. All bounds hold with probability $1 - \delta$, where the confidence parameter $0 < \delta < 1$ impacts the manner in which $k$ and $m$ must be set.

For the neural count sketch, we prove (Theorem 2) that frequencies upto a value $f$ are estimated within $\pm 1$ if the number of encoding neurons, $m = O(kn)$, and if the sparsity, $k = O(\max(n_f f^2) \log(1/\delta))$. For the 1-2-3-many sketch, we prove

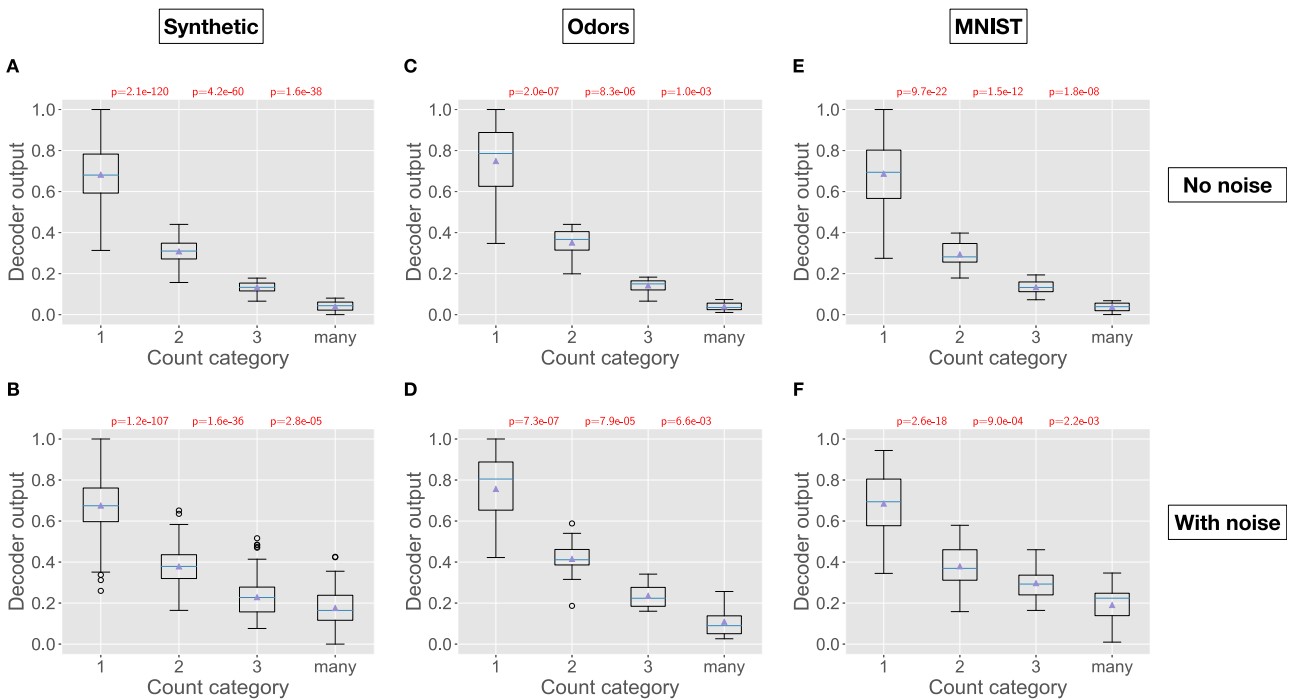

**Fig. 3 | Performance of the 1-2-3-many sketch (anti-Hebbian model).** In each panel, the x-axis shows the four ground-truth count categories, and the y-axis shows the decoder output from the 1-2-3-many sketch. For example, count category '1' includes all items being observed for the first time (i.e., items not present in the input sequence), and the decoder output shows the response magnitude of the decoder neuron for all such items. Boxplots show the median and first and third quartiles, with whiskers extending from the box by 1.5-times the inter-quartile range. Shown at the top are p-values (Wilcoxon rank-sum test, two-sided) comparing differences in decoder response magnitudes between items in successive count categories. For example, in panel A, the difference between decoder responses to items in count category '1' vs '2' was statistically significant, with $p = 2.1e\text{-}120$. All $p < 0.01$ are colored red. After removing correlated items, $N = 1000$ for Synthetic; $N = 62$ for Odors; and $N = 180$ for MNIST. Each panel shows a dataset (columns), with or without noise added to items (rows). **A, B** Synthetic without and with noise. **C, D** Odors without and with noise. **E, F** MNIST without and with noise.

(Theorem 5) that it is sufficient to have $m = O(kN)$ and $k = O(\log(1/\delta))$, which improves upon the neural count sketch in two important ways. First, the bound depends on the number of distinct items ($N$), rather than the total number of observations including repetitions ($n$), which could be far larger. Second, a significantly smaller setting of $k$ (and thus $m$) is sufficient. In other words, the 1-2-3-many sketch only needs a few synapses to be allocated per unique item to generate good estimates.

The superior performance of the 1-2-3-many sketch comes at the cost of a higher weight precision requirement. The count sketch can accurately report frequencies upto $f$ as long as its synaptic weights $w_j$ have $O(\log f)$ bits of precision. The 1-2-3-many sketch, on the other hand, needs $O(f)$ bits of precision per weight, which is still within empirical estimates for small $f$ (e.g., 3–5[54]).

We also look at what happens when items are not necessarily well-separated. In such situations, where items lie in a continuum without well-defined boundaries, the notion of frequency becomes murkier. In this setting, we show that, the count sketch functions as a kernel density estimate[67], where the sketch outputs a value that relates to the density of observations around a given item.

**Theoretical results for a special case.** The results above are proved in the Supplement (Notes 1–3) in a fairly general setting. For a concise illustration, consider the special case where the input vectors $x$ are of unit length and the random matrix $M$ has entries that are sampled independently from a standard normal distribution. Then Assumption 1, Theorem 2, and Theorem 5 take on the following form.

**Assumption 1′.** The $n$ observations seen by the sketch consist of $f_i$ repetitions of $x^{(i)}$, for $i = 1, 2, \dots N$, interleaved arbitrarily. For any $i \neq j$, we have $x^{(i)} \cdot x^{(j)} < \zeta$ for some constant $\zeta > 0$.

This says that the distinct observations are almost orthogonal, as would be expected if they were chosen independently at random from the unit sphere.

Theorem 2 gives two results for the neural count sketch: frequency estimates that are accurate within $\pm 1$ and looser estimates that are accurate within $\pm \epsilon n$.

**Theorem 2′.** There is an absolute constant $c$ for which the following holds. Suppose the neural count sketch sees $n$ observations satisfying Assumption 1′ with $\zeta \leq 1/(\log n)$. Pick any $0 < \delta < 1$.
- Suppose that $m \geq 2\,kn$ and that $k \geq c \max(n, f^2) \ln(1/\delta)$ for a positive integer $f$. Then with probability at least $1 - \delta$, when presented with a query $x^{(i)}$ with $0 \leq f_i \leq f$, the response of the neural count sketch will lie in the range $f_i \pm 1$.
- Suppose that $m \geq 2\,k/\epsilon$ for some $\epsilon > 0$ and that $k \geq (c/\epsilon^2) \ln(1/\delta)$. Then with probability at least $1 - \delta$, when presented with a query $x^{(i)}$, the response of the neural count sketch will lie in the range $f_i \pm \epsilon n$.

Note that the query $x^{(i)}$ need not belong to the original sequence of $n$ observations, in which case $f_i = 0$.

Theorem 5 gives bounds that are significantly more favorable for the 1-2-3-many sketch.

**Theorem 5′.** Suppose the 1-2-3-many sketch, with parameter $\beta = 1$, witnesses $n$ observations that satisfy Assumption 1′ with $\zeta \leq 1/(\log N)$. Pick any $0 < \delta < 1$ and suppose that $m \geq 2\,kN$ and $k \geq 12 \ln(2/\delta)$. Then with probability at least $1 - \delta$, when presented with a query $x^{(i)}$, the response of the sketch will be $e^{-r}$ for some value $r$ that is either $f_i$ or $f_i + 1$ when rounded to the nearest integer.

Overall, these mathematical proofs provide bounds on how accurately stimuli can be tracked using the two neural count sketches.

## The *Drosophila* mushroom body implements the anti-Hebbian count sketch

Here, we provide evidence supporting the "1-2-3-many" model from the olfactory system of the fruit fly, where circuit anatomy and physiology have been well-mapped at synaptic resolution[68,69]. The evidence described below includes the neural architecture of stimulus encoding, the plasticity induced at the encoding-decoding synapse, and the response precision of the decoding (counting) neuron. The latter two we derive from a re-analysis of data detailing novelty detection mechanisms in the fruit fly mushroom body[8], where odor memories are stored.

**Stimulus encoding (Fig. 4A).** In the fruit fly olfactory system[70], odors are initially represented by the firing rates of $d = 50$ types of odorant receptor neurons. After a series of pre-processing steps, including gain control[71,72], noise reduction[73], and divisive normalization[48,74], odors are represented by the firing rates of $d = 50$ types of projection neurons (PNs), which each receive input from sensory neurons expressing the same receptor type. Thus, an odor $x$ is a point in $\mathcal{R}_+^{50}$.

The first piece (assigning the odor a sparse, high-dimensional representation) is accomplished by 2000 Kenyon cells (KCs), which receive input from the PNs. Each KC samples randomly from approximately 6 of the 50 PN types[75] and sums up their firing rates. Hence, the random projection matrix $M$ is a sparse binary matrix, with about 6 ones per row. Next, each KC sends feed-forward excitation to an inhibitory neuron, called APL, which then sends feed-back inhibition to each KC. As a result, only the top 5% of highest-firing KCs remain active for the odor, and the rest are silenced[42,47,48]. Moreover, KCs tend to respond in a binary manner, firing either zero spikes or just a few spikes per odor[42,76,77]. Thus, odors are encoded as a high-dimensional binary vector (with dimension $m = 2000$), of which only a few KCs ($k = 100$) are active for the odor.

**Synapse weight updating (Fig. 4B, C).** The second piece involves synaptic connections from KCs to an output neuron. In the fly mushroom body, there are 35 types of output neurons (called MBONs[69,78]) that read-out information from the 2000 KCs and control behaviors, such as learning to approach or avoid odors[70]. KC → MBON synapses are plastic[79], and dopamine modulates the synaptic strength bidirectionally depending on the timing contingency between KC activity and dopamine release[8,80,81]. Synaptic changes are consistent with anti-Hebbian plasticity, albeit on a longer time scale than traditional STDP and without requiring post-synaptic firing[82].

Recently, one MBON (called MBON-$\alpha'3$) was discovered that computes the novelty of an odor[8] (Fig. 4B). When an odor is experienced, synapses from the odor's activated KCs onto MBON-$\alpha'3$ multiplicatively weaken, whereas synapses from non-active KCs onto MBON-$\alpha'3$ strengthen slightly ($\epsilon$ in Eq. (2)). The output of MBON-$\alpha'3$ is the weighted sum of its inputs (i.e., the activity of each KC multiplied by its synaptic strength). Thus, repeated exposure to the same odor depresses active KC → MBON-$\alpha'3$ synapses, which suppresses the activity of MBON-$\alpha'3$ in response to the odor, indicating that the odor has become familiar. Hattori et al.[8] also found another output neuron (called MBON-$\beta1 > \alpha$) that responds linearly with familiarity. Thus, this circuit uses repetition suppression (MBON-$\alpha'3$ for novelty) and possibly repetition enhancement (MBON-$\beta1 > \alpha$ for familiarity), though the latter remains unconfirmed mechanistically.

To quantify the weakening in the KC → MBON-$\alpha'3$ synaptic weights following stimulus experience, we re-analyzed MBON-$\alpha'3$ responses from 72 cells to 10 repeated exposures of the same odor (Fig. 4C). Each exposure increases the number of times the odor is experienced. The median normalized response of MBON-$\alpha'3$ to an odor experienced for the first time (category 1) was 1.00, compared to 0.413, 0.193, 0.098, and 0.048, for categories 2 through 5, respectively.

The data closely fit an exponential decay function ($R^2 = 0.996$), with a suppression constant of 0.44. This means that each successive exposure decays the MBON-$\alpha'3$ response by a factor of 0.44. Thus, $\beta = -\ln(0.44)$ in Eq. (2), supporting the general functional form of suppression proposed.

**Frequency decoding (Figure 4D–F).** While MBON-$\alpha'3$ was originally conceived as a binary novelty detector neuron[8], our re-analysis of MBON-$\alpha'3$ responses provides evidence for the presence of more than two discrete count categories along the novelty-familiarity axis. To show this, the activity of MBON-$\alpha'3$ must be significantly different across multiple experiences of the same odor. At some point, the difference in activity between successive experiences becomes indistinguishable, and this is where the "many" category kicks in, indicating that responses to all subsequent experiences are essentially the same. Specifically, for "count category" $j$ to exist, it must be possible to distinguish category $j$ from each other category, including each individual category encapsulated by "many".

Strikingly, re-analysis of MBON-$\alpha'3$ activity levels to successive experiences of an odor shows that the distinguishability of responses are consistent with the 1-2-3-many model (Fig. 4D). Categories 1, 2, and 3 were each significantly different from each other category (all $p < 0.01$; Wilcoxon rank-sum test). However, category 4 was not significantly different from categories 5 and 6, and categories $j = 5$ onwards were not significantly different from categories $j + 1$ onwards. Thus, the decoding neuron can robustly distinguish among odors experienced 1, 2, or 3-times before, with a separate category for 4 or more (many).

Visualization of the distributions of MBON-$\alpha'3$ responses to odors in each count category shows the separability of categories 1, 2, and 3, as well as the clustering of categories 4–10 (Fig. 4E). The blue curve (category 1) is clearly distinguishable from the orange curve (category 2), which is distinguishable from the red curve (category 3). However, the curves for categories 4 (green) and 5–10 (all black) are highly overlapping, indicating that their responses are roughly the same and comprise the 'many' category.

We also quantified the separability of all pairs of count categories using a simple response threshold discrimination model (Fig. 4F). The area under the ROC curve remained high ($\geq 0.70$) when discriminating between 1, 2, and 3 and nearly all other categories, but was considerably degraded for subsequent categories, further supporting the existence of four robust count categories.

These results suggest that MBON-$\alpha'3$ encodes frequency information about odor memories into four distinct categories along the novelty-familiarity axis.

## Discussion
### Summary
One role of theory in neuroscience is to propose plausible circuit mechanisms that support important neural computations. Here, we showed how a fundamental data structure used by computer scientists to count frequency events in streaming data could be implemented by canonical neural circuitry. This theory was supported by experimental data in the insect mushroom body, which gave credence to the 1-2-3-many count sketch, both qualitatively and quantitatively, in terms of the required neural architecture, the functional form of synaptic plasticity, and the output precision of the counting neuron.

Our proposed neural count sketch data structure has four properties: (i) it provides counts that are stimulus-specific; (ii) it has a large storage capacity, that is, it requires only a few synapses per unique item[18]; (iii) it offers robustness, that is, the ability to generalize counts across noisy versions of the same item; and (iv) it is fast and automatic, providing frequency estimates of inputs after two synapses of computation, requiring only tens to hundreds of milliseconds.

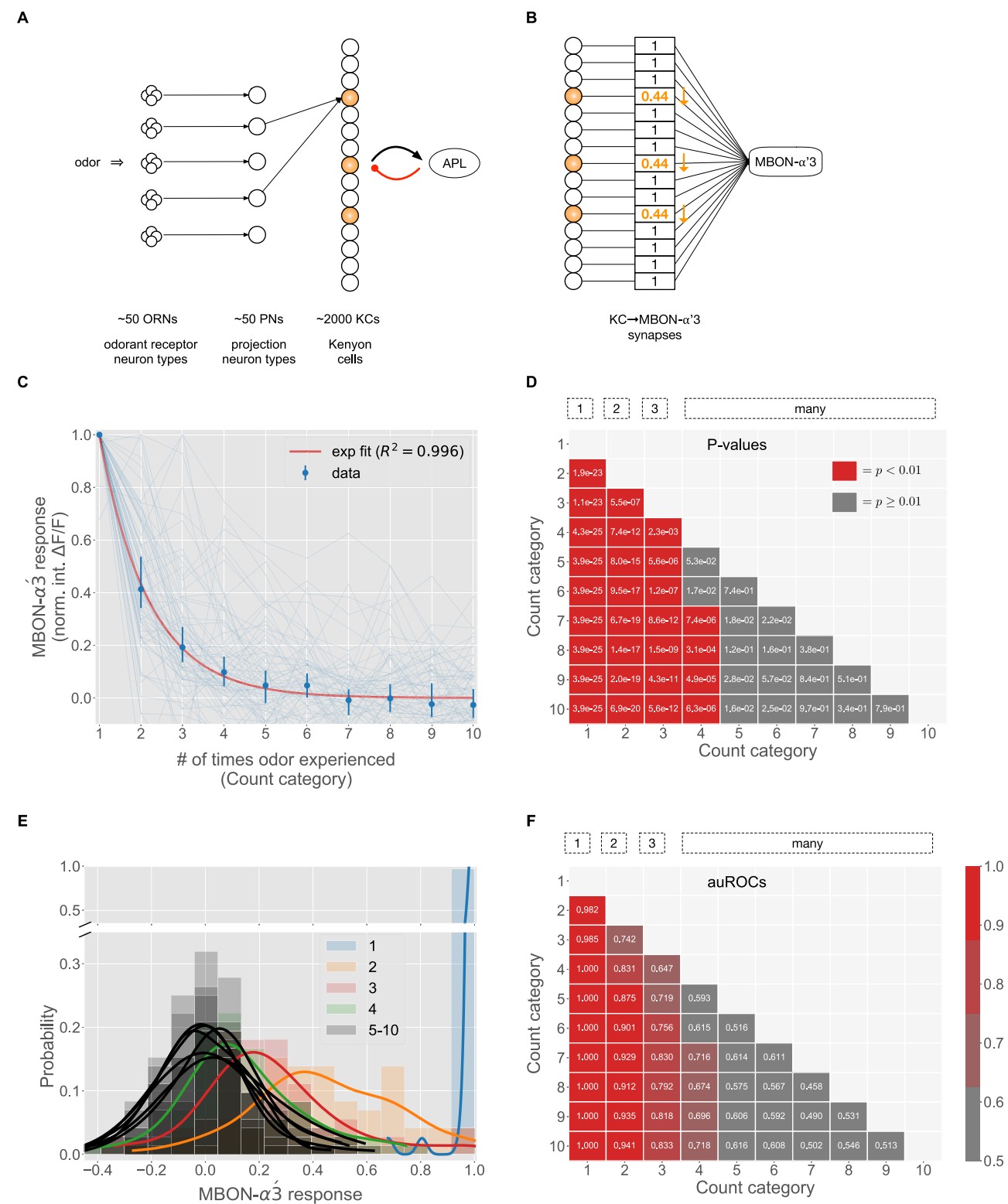

# Experimental questions and testable predictions

Our work raises several experimental and circuit design questions.

First, how might downstream mechanisms robustly read-out frequency estimates and use them to modify behavior? For the anti-Hebbian model, this would require grouping the firing rate of the 1-2-3-many counting neuron into four discrete categories. One option is to convert this continuous firing rate into a discrete (i.e., a "one-hot" encoded) representation (Fig. 5A). For example, the counting neuron

could synapse with four output neurons, each with successively lower firing thresholds and with inhibition from neurons with higher thresholds to neurons with lower thresholds. As a result, each count category will be represented by the activity of a single neuron. A second option is to hierarchically string together counting neurons (Fig. 5B). Here, one counting neuron inhibits the activity and synaptic plasticity of another counting neuron, such that the first neuron robustly encodes 1 and 2, and (after the inhibition from the first neuron

**Fig. 4 | Experimental evidence of the "1-2-3-many" sketch from the insect mushroom body. A** Schematic of the fruit fly olfactory system. Odors are initially represented by the firing rates of 50 odorant receptor neuron types, which send axons to 50 projection neuron (PN) types. PNs then send odor information to 2000 Kenyon cells (KCs), each of which provides feed-forward excitation to a large inhibitory neuron (called APL), which sparsifies the KC representation via feedback inhibition. **B** Synapses between activated KCs and the counting neuron (MBON-$\alpha'3$) are modified (weakened) when an odor is experienced. **C** Response dynamics of MBON-$\alpha'3$ (y-axis) over 10 successive presentations of odor MCH (x-axis). Data shows responses of $N = 72$ cells (light blue) over 59 flies. Blue dots (dark blue) show median response values, and error bars show 99% confidence intervals determined by 20,000 bootstraps. For each cell, responses are normalized to the magnitude of the first presentation. Red curve shows data fit to an exponential function ($y = ae^{bx}$), with a suppression constant of 0.44. Source data are provided as a Source Data file.

**D** Heatmap of p-values (Wilcoxon rank-sum test, two-sided) comparing differences in response magnitudes for all pairs of count categories. For example, MBON-$\alpha'3$ responses are significantly different comparing an odor seen for the first time vs. the second time ($p = 1.9e$-23), but responses are not significantly different comparing the 4th vs. the 5th experience ($p = 5.3e$-02). Red blocks indicate $p < 0.01$, and gray blocks indicate $p \geq 0.01$. **E** Histogram distributions of MBON-$\alpha'3$ responses for each count category, with kernel density estimation curves plotted on top. Categories 1, 2, 3, and 4 are shown in blue, orange, red, and green, respectively; categories 5–10 are shown in black. Categories 4–10 (many) are highly overlapping and distinguishable from categories 1, 2, and 3. **F** Heatmap of area under the ROC (auROC) values for discriminating between each pair of count categories using a linear threshold applied to the MBON-$\alpha'3$ response. For example, the auROC for discriminating 1-vs-2 is 0.982. Distinguishability is highest for categories 1, 2, and 3 (first three columns), and then tapers off for subsequent categories.

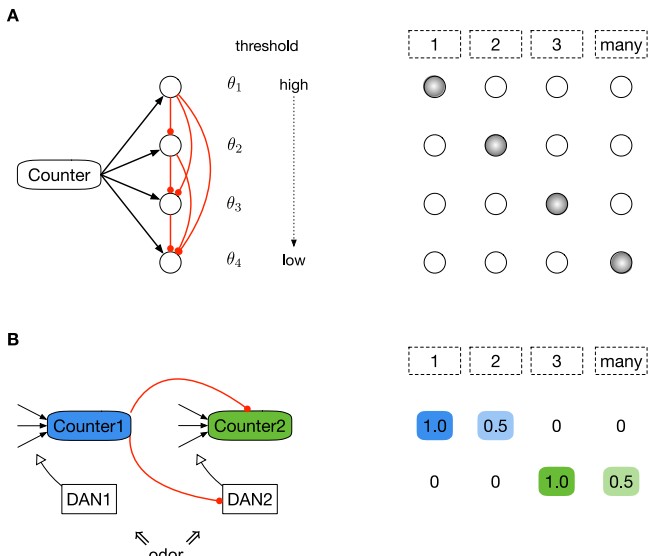

**Fig. 5 | Hypothetical read-out mechanisms of the counting neuron. A** This model translates the continuous firing rate of the 1-2-3-many counter neuron into a discrete representation. Because this model uses repetition suppression, larger firing rates correspond to less familiar odors. The counter synapses onto four downstream neurons, each with successively smaller firing thresholds ($\theta_1 > \theta_2 > \theta_3 > \theta_4$). Neurons with larger thresholds inhibit those with smaller thresholds. As a result, each count category becomes "one-hot" encoded, making it easier to modify behavior. **B** This model hierarchically strings together counting neurons to increase scalability and the resolution between count categories. In the insect mushroom body, odors activate dopamine neurons (DANs), which modulate synaptic weights onto counting neurons. In this model, there are two counting neurons with two associated DANs. Both counters receive input from encoding neurons, and Counter1 inhibits Counter2 and DAN2. Counter1 encodes categories 1 and 2 with high (1.0) and medium (0.5) responses, respectively. When Counter 1's activity is diminished after the second stimulus experience, the inhibition onto Counter 2 is lifted, allowing Counter2 to encode the two subsequent categories with high and medium responses.

corresponds to the response prior to the rise of the sigmoid, with a few categories in the middle, and then 'many' at the saturation of the sigmoid.

Second, our results suggest that behaviorally, animals can distinguish among stimuli in each of the four count categories, as opposed to just the traditional novel vs. familiar categorization. Ethologically, it seems important for organisms to discriminate between the first and second experience of a stimulus, since there are many things experienced once (e.g., randomly) but many fewer things experienced twice. Distinguishing between the second and third experiences may be advantageous during exploratory behavior. For example, an animal might enter and then leave a locale with some identifying scent, experiencing it twice, once upon entry and once more upon exit; returning again to the same locale could trigger a memory that the animal has already been there before. Similarly, another animal (say, a potential mate) may enter and then leave a locale, and knowing if that animal returns again could warrant a change in behavior. Indeed, many things come and go, but few things come back again. The final category hosts stimuli experienced 'many' times, indicative of re-occurring experiences that define one's environment (e.g., a mother's voice, the scent of a nest). It is also striking that some indigenous tribes only have words for "one", "two", "three", and "many"[83], which suggests that the value of having four distinct count categories may indeed be broadly conserved, even in humans.

Third, we analyzed the functional form of repetition suppression at single cell resolution, and we quantified how the setting of $\beta$ (the suppression constant) and other circuit parameters impact the distinguishability of count categories. How general is this form and the corresponding value of $\beta$ in the numerous other systems that use repetition suppression to encode stimulus familiarity[9,10,55–59]? Our theory also hypothesizes that count estimates are privy to the similarity structure of stimuli. For discrete, well-separated stimuli, our model predicts that animals can generalize counts across noisy versions of the same stimuli. For continuous stimuli, count estimates may reflect a kernel density estimate, capable of counting sub-features shared by stimuli.

Fourth, what are the factors, such as attention[84], arousal, and other brain states[80,85,86], that control whether counts are updated upon stimulus experience? In the mushroom body, repetition suppression occurs due to dopamine release in the $\alpha'3$ compartment after each experience of a stimulus. The lack of dopamine release may be indicative of an experience that is not "inserted" into the sketch and hence not remembered. This mechanism also provides the intriguing benefit of being able to query the count sketch for the frequency estimate of an item, without updating its count − i.e., a form of "recollection". In addition, the unit of "experience" that triggers dopamine release remains unclear. For images, is a single 2-second exposure equivalent to five successive exposures of 400ms each? For odors, what duration of an odor puff gets integrated into a single experience?

is lifted), the second neuron encodes 3 and many, etc. This option provides a mechanism to translate a small resolution counting system to a larger one, with greater separability between count categories. Thus, multiplexing counting modules via hierarchical connections could provide robustness and scalability.

For the Hebbian model, the read-out may simply be the total activity level, which scales with stimulus frequency. Indeed, in the mushroom body, the response of the familiarity neuron (MBON-$\beta1 > \alpha$[8]) increases linearly with successive odor experience, which supports the additive form of synaptic plasticity in Eq. (1). Alternatively, a discrete read-out could be generated by applying a sigmoid activation function to the counting neuron. Category 1 would

Fifth, what is the function of the many other "counting neurons" in the brain that track stimulus familiarity? One idea is that counts are conditioned on location; e.g., "how many times have we met in New York?" The hippocampus is believed to be a central location where counts and context may be integrated[2,9,87–89]. Another idea is that some neurons have faster or slower synaptic recovery rates ($\epsilon$), and thus, different memory spans. For example, in the insect mushroom body, different anatomical compartments acquire and forget memories at different rates, leading to short- and long-term memories[90]. For counting, non-zero values of $\epsilon$ provide a mechanism to free-up capacity for newer items at the expense of those not experienced in a while. This would also help prevent synapse saturation (to 1 for the Hebbian model, and to 0 for the anti-Hebbian model). Relatedly, there are variants of count sketches that allow for item deletion[91,92]. Thus, having multiple counting neurons can help contextualize frequency estimates across both space and time.

## Comparison to prior models

Earlier works (reviewed by Bogacz and Brown[18]) were pioneering in establishing plausible models for recognition memory. These models use three core computations that are also found in our model, albeit some important differences in how these computations are implemented. First, both models use sparse coding to represent stimuli; however, prior models assume the input feature vectors ($x$) are sparse and binary, where each neuron encodes a different feature, and the neuron is active if the corresponding feature is present in the stimulus. Our model assumes dense input vectors that represent stimuli using a combinatorial code[64]; we then apply a random expansion and winner-take-all competition to generate sparse, high-dimensional codes. Importantly, our mechanism is provably similarity-preserving[50,51], which allows counts to generalize across noisy versions of a stimulus. Second, both models store memories using Hebbian[93,94] or anti-Hebbian[9,95] plasticity. Our model, however, proposes a new version of the anti-Hebbian weight update − multiplicative LTD in Eq. (2) compared to subtractive LTD previously − which was an important determinant of the number of distinguishable count categories; i.e., multiplicative LTD creates larger separation between count categories compared to subtractive LTD, but it encodes fewer categories. Third, both models use decoder neurons that output stimulus familiarity. However, prior models only produce a binary output (is the stimulus novel or familiar?) whereas our model produces a graded output (level of familiarity). Our new anti-Hebbian rule, and the transition to a graded response, also required new forms of analysis to estimate the capacity of the models and, in our case, to bound its error. Finally, unlike prior models that were largely theoretical, our model was grounded in known anatomy and physiology from the *Drosophila* mushroom body, where inputs and outputs of encoding neurons, the sparsification mechanism, and the integration function of the novelty detection neuron are all precisely known.

There are also aspects of previous models that we did not take into account. First, our model only included one novelty detection neuron, whereas prior models included multiple novelty detection neurons that could detect novelty in the spatial domain[18,94]. For example, if neurons receive uncorrelated input, then different neurons could be used to identify which objects in a scene are novel, and which are not. In our model, this would be equivalent to identifying a novel component within an otherwise familiar odor mixture. We could incorporate this behavior into our future model by having multiple counting MBONs that sample from distinct Kenyon cells. Second, we assumed that stimulus representations ($z$) are static, whereas prior work also considers the case where representations change over time; e.g., familiar stimuli induce sparser and more precise representations than novel stimuli[15,16,55]. Third, Bogacz et al.[93] propose a conceptually different approach: using the energy function of the Hopfield network as an output of stimulus familiarity, where lower energy means the stimulus is more familiar. However, the neural correlate of this energy function has not been experimentally identified.

## Generality to other brain regions and species

There are two main ingredients of the neural count sketch data structures − sparse, high-dimensional representations for stimuli and repetition-based modulation of synaptic weights. Where else are these two features found in the brain? Sparse, high-dimensional representations are ubiquitous in sensory areas, such as in olfaction, vision, audition, and somatosensation, as well as in the hippocampus[39,96]. Some of these regions shape representations using decorrelation[97], sharpening[3,61], and pattern completion mechanisms, which would further boost the stimulus-specificity of counts. Repetition suppression has been observed in many mammalian brain regions, including the perirhinal cortex, prefrontal cortex, basal ganglia, and inferior temporal cortex, amongst others[9,10,98]. Repetition enhancement (e.g., familiarity neurons) have also been found in many of these regions[12,99], though less common. Thus, all the machinery required to implement count sketches are prevalent in the brain, and basic memory counting machinery may be broadly conserved.

## Applications to machine learning

How might neural count sketches be useful in machine learning applications? Two ideas come to mind. First, neural count sketches can be used to perform outlier detection, and thus, to modulate attention towards the most salient inputs. Traditional count sketches are only used to identify "heavy hitters" (i.e., very popular content), which constitute a small fraction of the observed items in a data stream. However, equally important are "light hitters", that is, items that are rare or have never been seen before, which may signal anomalies and require attention. The 1-2-3-many count sketch bridges these two extremes by providing fine resolution at the transition between novel and familiar, as well as a separate class ("many") for popular items. Second, neural count sketches can be used to guide exploratory search behavior in reinforcement learning applications. Exploring agents often only receive occasional feedback, such as a reward when food is found. During the majority of the times when feedback is not received, the novelty-familiarity spectrum can be supplemented as an intrinsic reward signal to drive exploration[1]. In other words, including a neural count sketch module within a reinforcement learning network would allow agents to use occurrence frequencies to adjust behavior away from highly familiar states and towards novel, less explored states, which may be more informative. More generally, pre-loading deep networks with computational modules for frequency estimation may be a useful component towards generalized decision-making[100].

## Reporting summary

Further information on research design is available in the Nature Research Reporting Summary linked to this article.

## Data availability

The MBON-$\alpha'$3 response data is provided in the Supplementary Information/Source Data file. Source data are provided with this paper.

## Code availability

All code is available at: https://github.com/metalloids/fly_counting.

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

## Acknowledgements

The authors thank Alison L. Barth, Tatiana Engel, David Freedman, Partha Mitra, Guruprasad Raghavan, Yang Shen, and Shyam Srinivasan for helpful discussions. S.N. was supported by the Pew Charitable Trusts, the NIDCD of the National Institutes of Health under award numbers 1R01DC017695 and 1UF1NS111692-01, and funding from the Simons Center for Quantitative Biology at Cold Spring Harbor Laboratory.

## Author contributions

S.D. and S.N. conceived, designed, and implemented the model. D.H. and S.N. analyzed the data. S.D. performed the theoretical analysis. All authors wrote the manuscript.

## Competing interests

The authors declare no competing interests.
