## [Peer Review File · Nature Communications]

A neural theory for counting memoriesREVIEWER COMMENTS

Reviewer #1 (Remarks to the Author):

The manuscript presents a neural model for how the number of times a specific stimulus has been encountered could be represented in the brain. Two variations of the model are presented: one that relies on Hebbian plasticity and outputs the frequency of occurrence of each stimulus, and one that uses anti-Hebbian plasticity to keep track of occurrence frequency only within the categories 1, 2, 3, and many. The model comprises a layer of encoding neurons that compute a sparse representation of input stimuli, and a decoding neuron that outputs a measure of occurrence frequency based on that representation. The models are shown to be robust in the estimation of occurrence frequency in three datasets, and their biological plausibility is demonstrated by mapping model components onto neuron types previously identified in the *Drosophila* mushroom body.

The manuscript is well-written and provides intriguing computational insights into the representation of occurrence frequency in the brain. The concrete mapping of model components onto neuronal types is very interesting and demonstrates the value and importance of such models. While the departure of the model from the binary novel-vs-familiar representation studied in previous models into a more fine-grained representation is interesting and novel, the manuscript could benefit from a more detailed discussion of previous work, specifically concerning how the model relates to previously published models.

Minor concerns:

1. The model presented in the manuscript is mechanistically very similar to previous models, e.g., in the use of a sparse representation layer combined with memorization using Hebbian/anti-Hebbian plasticity (Bogacz & Brown 2003). While the manuscript provides good arguments for the shift towards a more fine-grained representation of occurrence frequency than the binary one mostly used in previous work, it would be important to directly compare the proposed model to previous models and discuss how it is to be placed among them. For example, would discarding the binary decision neurons found in the output layer of the model presented in Bogacz et. al. 2001, and focusing instead on the intermediate “familiarity detection neurons” provide similar results to what is presented in the current manuscript?

2. The dimensionality of the standard MNIST dataset is $d=784$, not 84 (final paragraph of page 6).

Bogacz, R., & Brown, M. W. (2003). Comparison of computational models of familiarity discrimination in the perirhinal cortex. *Hippocampus*, 13(4), 494-524.

Bogacz, R., Brown, M. W., & Giraud-Carrier, C. (2001). Model of familiarity discrimination in the perirhinal cortex. *Journal of computational neuroscience*, 10(1), 5-23.

Reviewer #2 (Remarks to the Author):

Summary:

This paper proposes two simple neural mechanisms to implement counting of memories. The first counts hashed versions of all memories---classifying them into some number v of categories. The second reduces this to just four categories corresponding to 1-2-3-many. Both give approximately-accurate counts.

The designs of the mechanisms are an extension of prior work on novelty detection.

The mechanisms involve dimensionality expansion, followed by k -Winner-Take-All, and specially-tailored readout mechanisms.

Hebbian rules are used, and also anti-Hebbian rules for the 1-2-3-many case.

The authors carry out extensive experiments showing that their mechanisms do what they are supposed to. They also have what is essentially a complete theoretical paper in the Supplementary Information section, which proves mathematically that the mechanisms work.

The authors claim that versions of their mechanisms are consistent with previous experiments on fruit-flies.

They suggest many directions for future experimental work.

The results seem quite interesting, and the writing is generally high quality. I recommend that the paper be accepted to *Nature Communications*, subject to some fixes, below.

Suggestions:

p. 2:

You say repeatedly that you are solving a "continuous" version of the problem. This is to contrast your work with prior work that considered a binary version of the problem (previously seen or not). But I don't see anything continuous here, just a classification into v , or 4, categories. That is discrete. I suggest that you remove the "continuous" terminology when referring to these discrete classifications.

p. 2, line 11:

I think you mean "classical", not "classic" here. There is a difference.

p. 2, line -10:

Is "staple" the right word? Do you mean that it can be used for many purposes, or maybe as a sub-network for solving many tasks?

p. 2, line -1:

Do insects have "hardware"? That sounds a bit too metaphoric for a biology paper.

p. 3, line 6:

At this point, it is not clear what you mean by "approximate". Say more precisely?

p. 3, par. 2, line 1:

Start out by saying what k and v are: k is the number of hash functions and v is the number of count categories.

p. 3, par. 2, line 4:

I am mystified here. What exactly does it mean to say that the hash functions are "random"? What is the distribution from which they are chosen? You should define what you mean by "random" first, and then it will make sense to say that the hash functions are chosen independently from the common distribution.

I know, the distribution is inferrable from the detailed mechanism described later. But then it seems backwards to say that that mechanism is an "implementation" of a higher-level algorithmic strategy. You

should define the high-level algorithm carefully first, before saying that the neural network implements it.

p. 4:

At this point, I am not sure what "approximate" means. According to what measure? Deviation from the correct count?

p. 4, line 7:

You are using the fact that $k \ll v$ but I don't think you assumed this anywhere up to this point.

p. 8:

Perhaps more can be said in the main body of the paper about what is actually proved theoretically. Can you at least state the results precisely here, and reserve the proofs for the Supplementary Information section.

I did not go over the formal material in the Supplementary Information section carefully; I will rely on the authors for that.

p. 12:

Can you say a bit more about possible applications to Machine Learning?

Reviewer #1

1. *“The manuscript could benefit from a more detailed discussion of previous work, specifically concerning how the model relates to previously published models. ... The model presented in the manuscript is mechanistically very similar to previous models, e.g., in the use of a sparse representation layer combined with memorization using Hebbian/anti-Hebbian plasticity (Bogacz & Brown 2003). While the manuscript provides good arguments for the shift towards a more fine-grained representation of occurrence frequency than the binary one mostly used in previous work, it would be important to directly compare the proposed model to previous models and discuss how it is to be placed among them. For example, would discarding the binary decision neurons found in the output layer of the model presented in Bogacz et. al. 2001, and focusing instead on the intermediate “familiarity detection neurons” provide similar results to what is presented in the current manuscript?”*

We added the following section discussing how our model relates to previous models:

Comparison to prior models. Earlier works (reviewed by Bogacz and Brown 2003) were pioneering in establishing plausible models for recognition memory. These models use three core computations that are also found in our model, albeit some important differences in how these computations are implemented. First, both models use sparse coding to represent stimuli; however, prior models assume the input feature vectors (x) are sparse and binary, where each neuron encodes a different feature, and the neuron is active if the corresponding feature is present in the stimulus. Our model assumes dense input vectors that represent stimuli using a combinatorial code (Stevens, 2018); we then apply a random expansion and winner-take-all competition to generate sparse, high-dimensional codes. Importantly, our mechanism is provably similarity-preserving (Dasgupta et al., 2017, 2018), which allows counts to generalize across noisy versions of a stimulus. Second, both models store memories using Hebbian (Bogacz et al., 1999, 2001) or anti-Hebbian (Bogacz and Brown, 2002; Brown and Xiang 1998) plasticity. Our model, however, proposes a new version of the anti-Hebbian weight update — multiplicative LTD in Equation (2) compared to subtractive LTD previously [e.g., see Equation (A.6) in Bogacz and Brown (2003)] — which was an important determinant of the number of distinguishable count categories; i.e., multiplicative LTD creates larger separation between count categories compared to subtractive LTD, but it encodes fewer categories. Third, both models use decoder neurons that output stimulus familiarity. However, prior models only produce a binary output (is the stimulus novel or familiar?) whereas our model produces a graded output (level of familiarity). Our new anti-Hebbian rule, and the transition to a graded response, also required new forms of analysis to estimate the capacity of the models and, in our case, to bound its error. Finally, unlike prior models that were largely theoretical, our model was grounded in known anatomy and physiology from the *Drosophila* mushroom body, where inputs and outputs of encoding neurons, the sparsification mechanism, and the integration function of the novelty detection neuron are all precisely known.

There are also aspects of previous models that we did not take into account. First, our model only included one novelty detection neuron, whereas prior models included multiple novelty detection neurons that could detect novelty in the spatial domain (Bogacz et al., 2001,

2003). For example, if neurons receive uncorrelated input, then different neurons could be used to identify which objects in a scene are novel, and which are not. In our model, this would be equivalent to identifying a novel component within an otherwise familiar odor mixture. We could incorporate this behavior into our future model by having multiple counting MBONs that sample from distinct Kenyon cells. Second, we assumed that stimulus representations (z) are static, whereas prior work also considers the case where representations change over time; e.g., familiar stimuli induce sparser and more precise representations than novel stimuli (Li et al. 1993; Norman and O'Reilly 2001; Sohal and Hasselmo 2000). Third, Bogacz et al. (1999) propose a conceptually different approach: using the energy function of the Hopfield network as an output of stimulus familiarity, where lower energy means the stimulus is more familiar. However, the neural correlate of this energy function has not been experimentally identified.

The familiarity detection neurons (FDNs) of Bogacz et al. (2001) do receive graded inputs, which are then binarized into a novel or familiar output. The plasticity rule used between inputs and FDNs is similar to what we propose: a Hebbian update that adds a constant to the weight of active inputs. However, no sparse encoding mechanism is proposed (i.e., the transformation from x to z), and thus it is difficult to compare directly to their work. Moreover, Bogacz et al. (2001) do not propose an anti-Hebbian rule.

2. The dimensionality of the standard MNIST dataset is $d=784$, not 84 (final paragraph of page 6).

We apologize for the confusion. While the input dimension of the standard MNIST dataset is indeed 784, we first applied a pre-processing step to extract discriminative features of the input images. These details were described in the Supplement (with some slight modifications now in red): “The third dataset, MNIST, consists of $N=10000$ images of handwritten digits. Because these raw images consist largely of black pixels, the similarity between many pairs of images, **regardless of their class**, will be quite high. So, instead of using the raw pixel representation, we trained a LeNet5 network using the 10 class labels. We then extracted a $d=84$ dimensional feature representation of each image **from the inner-most hidden layer in the LeNet5 network. This representation better captured the true similarity structure of digits and resulted in less count interference, compared to the raw input.**”

To make this clearer in the main body, we also added: “... where each image is of dimension $d=84$ **(after applying a pre-processing step to extract discriminative features; Supplement).**”

Reviewer #2

1. *"p. 2: You say repeatedly that you are solving a "continuous" version of the problem. This is to contrast your work with prior work that considered a binary version of the problem (previously seen or not). But I don't see anything continuous here, just a classification into v, or 4, categories. That is discrete. I suggest that you remove the "continuous" terminology when referring to these discrete classifications."*

The output of the counting neuron is continuous; it is just a firing rate equal to the sum of activities times weights (see the equation in the section "Frequency decoding"). However, the correctness of the firing rate is based on how close it is to the true integer frequency of the stimulus (for the Hebbian model) or the true count category (anti-Hebbian). In both cases, the desired output indeed takes on one of several possible discrete values, as you said. In addition, for the anti-Hebbian model, we proposed mechanisms to convert the continuous firing rate to a discrete output (Figure 5), and our theoretical analysis relies on discretizing the continuous output to assess its counting ability.

So, we agree that the term "continuous" is vague and potentially misleading. To address this, throughout the manuscript, we changed "continuous" to say "**several possible discrete classes**" (or something to a similar effect).

2. *"p. 2, line 11: I think you mean "classical", not "classic" here. There is a difference."*

Changed.

3. *"p. 2, line -10: Is "staple" the right word? Do you mean that it can be used for many purposes, or maybe as a sub-network for solving many tasks?"*

We meant that it's a general architecture commonly found in the brain. To avoid confusion, we deleted the word "staple".

4. *"p. 2, line -1: Do insects have "hardware"? That sounds a bit too metaphoric for a biology paper."*

Changed to "**circuitry**".

5. *"p. 3, line 6: At this point, it is not clear what you mean by "approximate". Say more precisely?"*

Changed to: “A count sketch is a data structure that **outputs counts for an item which are approximately equal to the true counts of the item, ...**”

6. *“p. 3, par. 2, line 1: Start out by saying what k and v are: k is the number of hash functions and v is the number of count categories.”*

Added: **“... where k is the number of hash functions, and v is the range of the hash functions.”**

7. *“p. 3, par. 2, line 4: I am mystified here. What exactly does it mean to say that the hash functions are “random”? What is the distribution from which they are chosen? You should define what you mean by “random” first, and then it will make sense to say that the hash functions are chosen independently from the common distribution. ... I know, the distribution is inferrable from the detailed mechanism described later. But then it seems backwards to say that that mechanism is an “implementation” of a higher-level algorithmic strategy. You should define the high-level algorithm carefully first, before saying that the neural network implements it.”*

To explain what we mean by a “random” hash function, we added: **“In a “random” hash function, every column index in the output range is generated with roughly the same probability. This means that the inputs are spread uniformly over the range, and two similar inputs could be assigned to arbitrarily far apart indices.”**

This becomes important later when we contrast random hash functions with locality-sensitive hash functions, which would assign two similar inputs to nearby indices, with high probability.

8. *“p. 4: At this point, I am not sure what “approximate” means. According to what measure? Deviation from the correct count?”*

Changed to: **“Thus, the level of approximation (i.e., the amount of deviation from the correct count) depends on the amount of overlap with other items, as well as ...”**

9. *“p. 4, line 7: You are using the fact that $k \ll v$ but I don't think you assumed this anywhere up to this point.”*

Sorry for the confusion. We added the following to the end of the previous section: **“Typically, v is set to be much larger than k since v relates to the error of the count estimate for each hash function, and k simply averages these errors over multiple hash functions.”**

10. “p. 8: Perhaps more can be said in the main body of the paper about what is actually proved theoretically. Can you at least state the results precisely here, and reserve the proofs for the Supplementary Information section.”

Because it is difficult to succinctly state the full theorems, we added to the main text theorem statements for a special case where the inputs vectors (x) are of unit length and the random matrix (M) is sampled independently from a standard normal distribution. In this case, we precisely state the equivalent of Assumption 1 (inputs are nearly orthogonal to each other), Theorem 2 (analysis of the neural count sketch), and Theorem 5 (analysis of the 1-2-3-many count sketch). The more general theorem statements and full proofs are in the Supplement. **Please see the red text on pages 8-9 of the main text for details of these additions.**

11. “p. 12: Can you say a bit more about possible applications to Machine Learning?”

We expanded this section as follows: “How might neural count sketches be useful in machine learning applications? Two ideas come to mind. First, neural count sketches can be used to perform outlier detection, and thus, to modulate attention towards the most salient inputs. Traditional count sketches are only used to identify “heavy hitters” (i.e., very popular content), which constitute a small fraction of the observed items in a data stream. However, equally important are “light hitters”, that is, items that are rare or have never been seen before, which may signal anomalies and require attention. The 1-2-3-many count sketch bridges these two extremes by providing fine resolution at the transition between novel and familiar, as well as a separate class (“many”) for popular items. Second, neural count sketches can be used to guide exploratory search behavior in reinforcement learning applications. Exploring agents often only receive occasional feedback, such as a reward when food is found. During the majority of the times when feedback is not received, the novelty-familiarity spectrum can be supplemented as an intrinsic reward signal to drive exploration (Jaegle et al., 2019). In other words, including a neural count sketch module within a reinforcement learning network would allow agents to use occurrence frequencies to adjust behavior away from highly familiar states and towards novel, less explored states, which may be more informative. More generally, pre-loading deep networks with computational modules for frequency estimation may be a useful component towards generalized decision-making (Nasr et al., 2019).”

REVIEWERS' COMMENTS

Reviewer #1 (Remarks to the Author):

I think the paper is fine now. The detailed comparison to prior work addresses the main concern I had.

Reviewer #2 (Remarks to the Author):

The revisions look fine. I think the paper is now ready for publication.